# Clinical Predictors of Mortality in Prehospital Distress Calls by Emergency Medical Service Subscribers

**DOI:** 10.3390/jcm10225355

**Published:** 2021-11-17

**Authors:** Gabby Elbaz-Greener, Shemy Carasso, Elad Maor, Lior Gallimidi, Merav Yarkoni, Harindra C. Wijeysundera, Yitzhak Abend, Yinon Dagan, Amir Lerman, Offer Amir

**Affiliations:** 1Hadassah Medical Center, Cardiology Department, Faculty of Medicine, Hebrew University Jerusalem, Jerusalem 91905, Israel; myarkoni@hadassah.org.il (M.Y.); oamir@hadassah.org.il (O.A.); 2Baruch-Pade Poriya Medical Center, Cardiology Department, Azrieli Faculty of Medicine in the Galilee, Bar-Ilan University, Safed 52100, Israel; shemy.carasso@gmail.com; 3Leviev Heart Center, Sheba Medical Center and Sackler School of Medicine, Tel Aviv University, Tel-Aviv 69978, Israel; eladmaor@gmail.com; 4SHL Telemedicine Ltd., Tel-Aviv 67891, Israel; liorga@shahal.co.il (L.G.); Yitzhaka@shahal.co.il (Y.A.); yonid@shahal.co.il (Y.D.); lerman.amir@mayo.edu (A.L.); 5Schulich Heart Centre, Division of Cardiology, Sunnybrook Health Sciences Centre, University of Toronto, Toronto, ON M5S 1A1, Canada; harindra.wijeysundera@sunnybrook.ca; 6Department of Cardiovascular Disease, Mayo Clinic, Rochester, MN 55902, USA

**Keywords:** prehospital mortality, octogenarians, hypertension paradox, outcome

## Abstract

(1) Introduction: Most studies rely on in-hospital data to predict cardiovascular risk and do not include prehospital information that is substantially important for early decision making. The aim of the study was to define clinical parameters in the prehospital setting, which may affect clinical outcomes. (2) Methods: In this population-based study, we performed a retrospective analysis of emergency calls that were made by patients to the largest private emergency medical services (EMS) in Israel, SHL Telemedicine Ltd., who were treated on-site by the EMS team. Demographics, clinical characteristics, and clinical outcomes were analyzed. Mortality was evaluated at three time points: 1, 3, and 12 months’ follow-up. The first EMS prehospital measurements of the systolic blood pressure (SBP) were recorded and analyzed. Logistic regression analyses were performed. (3) Results: A total of 64,320 emergency calls were included with a follow-up of 12 months post index EMS call. Fifty-five percent of patients were men and the mean age was 70.2 ± 13.1 years. During follow-up of 12 months, 7.6% of patients died. Age above 80 years (OR 3.34; 95% CI 3.03–3.69, *p* < 0.005), first EMS SBP ≤ 130 mm Hg (OR 2.61; 95% CI 2.36–2.88, *p* < 0.005), dyspnea at presentation (OR 2.55; 95% CI 2.29–2.83, *p* < 0001), and chest pain with ischemic ECG changes (OR 1.95; 95% CI 1.71–2.23, *p* < 0.001) were the highest predictors of 1 month mortality and remained so for mortality at 3 and 12 months. In contrast, history of hypertension and first EMS prehospital SBP ≥ 160 mm Hg were significantly associated with decreased mortality at 1, 3 and 12 months. (4) Conclusions: We identified risk predictors for all-cause mortality in a large cohort of patients during prehospital EMS calls. Age over 80 years, first EMS-documented prehospital SBP < 130 mm Hg, and dyspnea at presentation were the most profound risk predictors for short- and long-term mortality. The current study demonstrates that in prehospital EMS call settings, several parameters can be used to improve prioritization and management of high-risk patients.

## 1. Introduction

The prehospital environment provides immediate and essential care for patients with life-threatening cardiovascular conditions such as cardiovascular death, myocardial infarction, heart failure (HF), exacerbation, or stroke. In addition to prompt treatment, prehospital care also provides vital patient information for the in-hospital medical team to improve and prioritize preferred admission.

Patients requiring prehospital care are generally those who acquire an emergency medical services (EMS) plan, essential for the patients’ health and well-being. As such, these patients do not represent the general population’s medical background, but rather a sector with high cardiovascular risk, high socioeconomic status, and prior comorbid conditions or previous hospitalizations, and signify the scope of this paper. For these patients, fully disclosed information, including all medications, is readily available to the dispatch team and hospital staff.

Studies investigating prehospital patient data, including early predictors for mortality, are becoming more common in recent years, and prior publications have reported multiple risk predictors for use in the prehospital occurrence for numerous conditions, including HF, diabetes mellitus (DM), and hypertension (HTN) [1]. Currently, there is considerable competing information regarding which prediction models should be used or recommended [2,3].

Clinical risk prediction measures optimize the decision making of medical staff. Patients at lower risk may be managed with less demanding care upon hospitalization, whereas patients at higher risk may require more thorough management and monitoring in an intensive care unit. Most studies based their information on predictors of mortality applicable to in-hospital patients and did not include the prehospital information that is substantially important for early decision making.

Risk models including prehospital information such as time from symptom onset to first medical care may provide additional value in the assessment of the patient’s condition and risks, which may be imperative for improving prognosis, as has been shown for acute coronary syndrome patients [4] and HF patients [5].

The current study was designed to assess and define the real-world clinical parameters in the prehospital setting and to evaluate their effect on patients’ outcomes.

## 2. Methods

### 2.1. Data Collection

This retrospective study was approved by the Institutional Review Board of the Baruch Padeh Medical Center (Code ID 0099-14-POR).

### 2.2. Data Source

Our study utilized data collected from the EMS–SHL database. SHL Telemedicine Ltd. ( Tel-Aviv, Israel) is the largest private telemedicine medical service in Israel. As mentioned above, EMS–SHL data represent a selected population of patients, from a particular socioeconomic background, usually having prior comorbidities and previous hospitalizations. The SHL database contains demographics, comorbidities, and clinical variables from the emergency scene, such as systolic blood pressure (SBP) taken and documented on-site upon arrival by the EMS team, which were all recorded and analyzed for the study.

The medical background for each patient was derived from a specific computerized individual SHL medical record, which was based on and validated by combined SHL personal medical history questionnaires and available hospital and medical documentation.

### 2.3. Patient Selection

A total of 64,320 emergency calls ended with SHL EMS team dispatch (considering the first dispatch for each subscriber) were made by patients to SHL between the years 1988 and 2018. These patients were treated on-site by the EMS team (physicians and paramedics), followed by further evacuation of the patients by ambulance to the nearest hospital, or the patients remained at home. Patients were followed, per the study’s retrospective protocol, for up to 12 months post index EMS call. Patients who needed cardiopulmonary resuscitation prior to or upon EMS arrival were excluded from the study.

### 2.4. Demographic and Clinical Measurement

A total of nine characteristics in the study were assessed: (1) age; (2) sex; (3) history of congestive heart failure (CHF); (4) history of HTN; (5) history of DM; (6) history of ischemic heart disease (IHD); (7) SBP as performed and documented by the EMS team, prehospital, and upon first contact on-site; (8) chest pain with ischemic ECG changes at presentation (ST-segment elevation and ST-segment depression); and (9) dyspnea at presentation.

The patient population was divided according to SBP taken by the EMS on-site at the time of arrival, and was divided accordingly into three tertiles: SBP < 130 mm Hg, SBP between 130 and 159 mm Hg, and SBP ≥ 160 mm Hg.

### 2.5. Outcome Variables

Our primary clinical outcome was mortality rate, which was recorded at 1, 3, and 12 months, following the initial index call to the center and upon EMS arrival. Mortality was defined as death from any cause from the day of the initial index call to the center and within the first year post-call. All-cause mortality was determined retrospectively for all patients from SHL Telemedicine medical records and by matching identification numbers of patients with the Israeli national population registry.

### 2.6. Statistical Analysis

Continuous variables are presented as mean ± standard deviation (SD) and categorical variables are presented as frequency and percentage. Age is also represented as a binary categorical variable equal/below and above 80 years old.

Univariate and multivariate logistic regression analyses were performed on the whole population of patient calls to identify parameters associated with mortality at 1, 3, and 12 months post the initial index call. All covariates whose univariate statistical significance was *p* < 0.05 were forced into a multivariable model. Odds ratio (OR) with a 95% confident interval (CI) and *p*-values were derived from the Wald chi-square test. A *p*-value of <0.05 was considered statistically significant. All statistical analyses were performed using the logistic regression analysis, in MedCalc^®^ statistical software version 20.014 (MedCalc Software Ltd., Ostend, Belgium; https://www.medcalc.org; accessed on 17 May 2021).

## 3. Results

### 3.1. Study Cohort

A total of 64,320 patients were included in the study between the years 1988 and 2018 and completed a 12 month follow-up. Baseline characteristics of the total cohort are represented in Table 1. The average age was 70 ± 13 years, and 35,316 (54.9%) of the patients were men.

A total of 35,087 (54.6%) patients had HTN, 31,257 (48.6%) patients had IHD, 13,897 (21.6%) patients had DM, and 10,798 (16.8%) had CHF (Table 1).

Of all 64,320 patients, 1599 (2.5%) died after 1 month, 2549 (4.0%) died after 3 months, and 4898 (7.6%) died after 12 months post prehospital EMS contact.

### 3.2. Univariate Analysis

The results from the univariate analysis are presented in Table 2. Age above 80 years was the strongest predictors of mortality at 1, 3, and 12 months (OR 3.34, 95% CI 3.0–3.7, *p*-value < 0.005; OR 3.21, 95% CI 3.0–3.5, *p*-value < 0.005; and OR 3.06, 95% CI 2.9–3.3, *p*-value < 0.005, respectively). Other comorbidities found to be significant negative predictors of mortality at 1, 3, and 12 months were history of CHF and male patients. IHD and DM were associated significantly with 1 year mortality, but not with early mortality 1–3 months after the index call (Table 2). Dyspnea at presentation was a stronger predictor of mortality compared to chest pain with ischemic ECG changes at presentation (Table 2).

On the contrary, prior HTN was the only comorbidity that was associated with lower mortality throughout the 1 year of follow-up with an OR of 0.62 (95% CI 0.6–0.7, *p* < 0.005; OR of 0.7 (95% CI 0.65–0.75, *p* < 0.005), and OR 0.81 (95% CI 0.8–0.9, *p* < 0.005) at 1, 3 and 12 months, respectively (Table 2).

Prehospital first SBP measurement was found to be an important predictor of outcome. First EMS SBP < 130 mm Hg was a strong predictor of mortality at 1 month with an OR of 2.6 (95% CI 2.4–2.9, *p* < 0.005); at 3 months, an OR of 2.34 (95% CI 2.2–2.5, *p* < 0.005); and at 1 year, an OR of 1.96 (95% CI 1.9–2.1, *p* < 0.005). In contrast, first SBP ≥ 160 mm Hg was the strongest predictor associated with reduced mortality at 1 month with an OR of 0.56 (95% CI 0.5–0.6, *p* < 0.005) at 3 months with and OR of 0.61 (95% CI 0.6–0.7, *p* < 0.005), and at 1 year with an OR of 0.66; (95% CI 0.6–0.7, *p* < 0.005) (Table 2).

### 3.3. Multivariable Logistic Analysis

The results of the multivariate logistic analysis are presented in Table 3. Similar to the univariate results, age above 80 years, first EMS SBP < 130 mm Hg, male sex, and CHF were significant negative predictors of mortality while prior HTN and first SBP ≥ 160 mm Hg were significant protective predictors. Dyspnea and chest pain with ischemic ECG changes as presentation symptoms were significant negative predictors of mortality at 1, 3, and 12 months.

Patients who had chest pain without ischemic ECG changes had better a survival rate compared to patients who had chest pain with ischemic ECG changes (Figure 1). We found a linear relationship between survival rate and SBP, with higher survival rate as SBP increased (Figure 2).

## 4. Discussion

Our study was designed on parameters that affect mortality in a prehospital setting of 64,000 on-site-treated patients who required all-comers EMS aid covering data for three decades between the years 1988 and 2018.

The current study demonstrates that age of over 80 years and first prehospital EMS SBP < 130 mm Hg were the parameters mot strongly associated with increased mortality.

Prior studies that assessed prehospital predictors of mortality demonstrated similar results: older age and low SBP [4,5,6,7,8,9,10] significantly increased mortality rates. In certain studies, increasing age [8] or low SBP [10] was indexed as part of an overall score of prehospital determinants that later was later used to assess mortality in the prehospital setting. However, these studies were mainly performed in specific populations such as in patients that were admitted to the emergency department following sudden cardiac arrest [4,5,11].or acute coronary syndrome [7].

Most studies rely on in-hospital data to predict cardiovascular risk. While few studies suggested that older age and SBP < 100–115 mm Hg were associated with higher mortality rate, they usually looked at a specific subgroup, mostly HF, and in an emergency room setting or in hospital [1,12,13,14,15,16,17,18,19]. Furthermore, older age and comorbidities were associated negatively with survival rates, both in emergency care and in-hospital [1,19,20].

Our data presented herein are of all-cause-mortality, regardless of the type of medical history. The study results demonstrated that age > 80 years and first EMS SBP < 130 mm Hg have a significant negative effect on mortality, similarly in the prehospital setting. When we addressed the significance of the first documented prehospital SBP measured by the EMS team upon arrival, we found that being in the lower tertile (<130 mm Hg) was an independent risk factor for short- and long-term mortality independent of age, sex, and other comorbidities.

In contrast, a relatively high SBP was noted in a prior study as protective against all-cause mortality in acute decompensation HF [17]. In chronic HF patients, a 13% decrease in mortality was observed per 10 mm Hg increase in SBP [21]. Similar observations were noted in HF with preserved and mainly with reduced ejection fraction, being termed the HTN paradox [12,17,19,22,23,24,25,26]. We showed that previous history of HTN and first prehospital EMS SBP ≥ 160 mm Hg were associated with reduced mortality at 1, 3, and 12 month follow-ups. Prior HTN and first SBP of ≥160 mm Hg measured by the EMS team were protective factors of mortality independent of age and sex.

The current study may have important implications on the prioritization and identification of high-risk individuals for early intervention. The most important information is that older age, previous medical history, and deviation from normal blood pressure parameters were independently associated with mortality. Furthermore, chest pain with ischemic ECG changes was a stronger predictor of mortality compared to chest pain without ischemic ECG changes. These factors are important to consider in early risk assessment. The development of a decision support tool may increase the possibility to differentiate patients with high-risk features from those without at an early stage.

The findings of this study should be considered in the context of several limitations. While the large amounts of data make our analysis much more robust, our data were obtained in a retrospective study, and as such, a limitation of unmeasured factors may have influenced the clinical outcomes, such as medication before and interventions during hospitalization. Furthermore, no data about other risk factors such as CKD, obesity, and smoking status were included in our study, despite increasing interest. Additionally, there was a population bias. Our study looked at a particular group of patients: those who could afford a paid emergency plan such as EMS-SHL, and who were regarded as at risk for particular medical conditions, thus requiring a plan. Our study excluded the population of patients potentially admitted to hospital due to life-threatening cardiovascular conditions via other transport ways.

Our study is novel in that it covers a large heterogeneous real-life patient population in the prehospital setting of emergency care, assessing both short- and long-term mortality. The ability to predict risk early on at initial stages such as prehospital allows medical staff to offer the best care, treatment, and therapy for each patient.

In conclusion, we identified risk predictors for all-cause mortality in a large cohort of patients during prehospital EMS calls. Age over 80 years, first EMS-documented prehospital SBP < 130 mm Hg, and dyspnea at presentation were the most profound risk predictors for short- and long-term mortality, while a history of hypertension and first EMS-documented pre-hospital SBP ≥ 160 mm Hg were protective for survival. A linear relationship was found between SBP and survival rate.

## Figures and Tables

**Figure 1 jcm-10-05355-f001:**
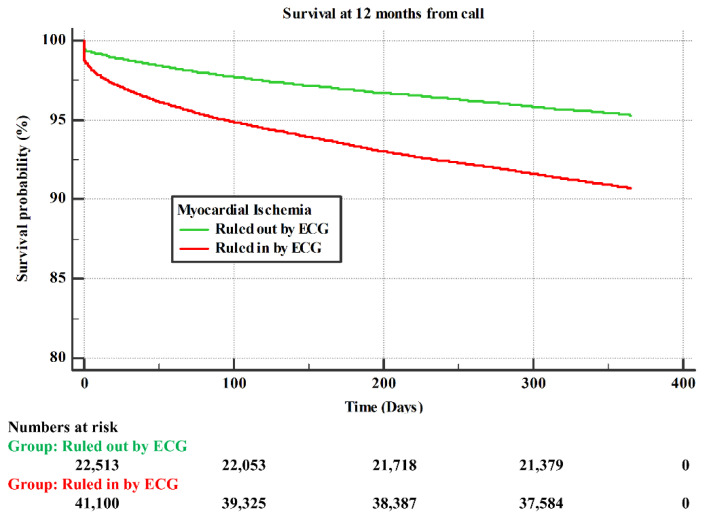
Survival rates at 12 months post index EMS call. Represented as a function of time (in months) for a population of patients that were ruled out by ECG for myocardial ischemia (green), and for patients that were ruled in by ischemic ECG changes (red).

**Figure 2 jcm-10-05355-f002:**
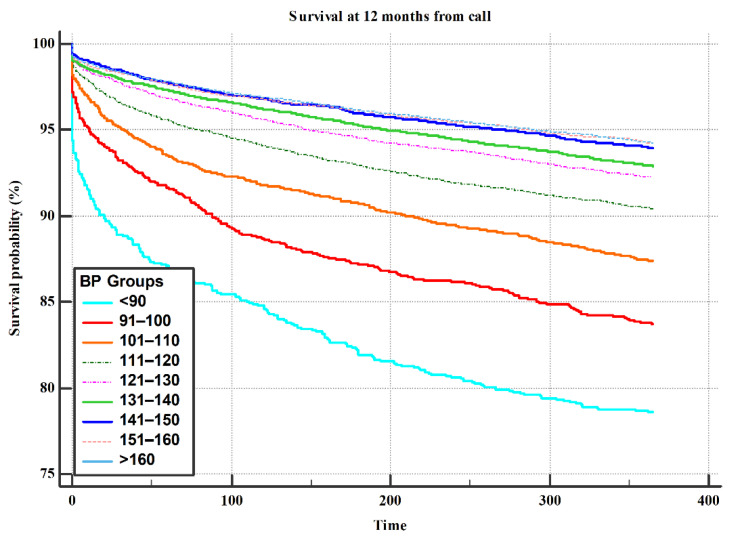
Survival rates at 12 months post index EMS call via SBP in increments of 10 mm Hg.

**Table 1 jcm-10-05355-t001:** Prehospital baseline characteristics.

Characteristic	
Age, years (*n* ± SD)	70.2 ± 13.1
*n* (%)	
Age > 80 years	13,687 (21.3)
Male	35,316 (54.9)
Congestive Heart Failure	10,798 (16.8)
Hypertension	35,087 (54.6)
Diabetes Mellitus	13,897 (21.6)
Ischemic Heart Disease	31,257 (48.6)

**Table 2 jcm-10-05355-t002:** Prehospital baseline characteristics and mortality–univariate analysis.

Characteristic Total *n* = 64,320	1 Month Mortality	3 Month Mortality	12 Month Mortality
Mortality, *n*, (%)	1599, (2.5)	2549, (4.0)	4898, (7.6)
	OR (95% CI), *p*-value	OR (95% CI), *p*-value	OR (95% CI), *p*-value
Age over 80	3.34 (3.03–3.69), <0.01	3.21 (2.97–3.47), <0.01	3.06 (2.89–3.24), <0.01
Male	1.21 (1.09–1.34), <0.01	1.23 (1.14–1.33), <0.01	1.28 (1.21–1.36), <0.01
Comorbidities			
Congestive Heart Failure	1.45 (1.29–1.63), <0.01	1.63 (1.49–1.79), <0.01	2.05 (1.93–2.18), <0.01
Prior Hypertension	0.62 (0.56–0.69), <0.01	0.7 (0.65–0.75), <0.01	0.81 (0.77–0.86), <0.01
Diabetes Mellitus	0.98 (0.87–1.11), 0.78	1.02 (0.93–1.12), 0.73	1.18 (1.11–1.26), <0.01
Ischemic Heart Disease	0.91 (0.83–1), 0.61	1.01 (0.93–1.09), 0.90	1.17 (1.1–1.24), <0.01
SBP in First Medical Contact			
First EMS SBP < 130 mm Hg	2.61 (2.36–2.88), <0.01	2.34 (2.16–2.53), <0.01	1.96 (1.85–2.07), <0.01
First EMS SBP ≥ 160 mm Hg	0.56 (0.5–0.63), <0.01	0.61 (0.56–0.67), <0.01	0.66 (0.62–0.71), <0.01
Main symptom at Presentation			
Chest Pain with ECG Ischemic Changes	2.53 (2.23–2.88), <0.01	2.35 (2.12–2.59), <0.01	2.05 (1.93–2.23), <0.01
Dyspnea	2.84 (2.60–3.19), <0.01	2.90 (2.67–1.15), <0.01	2.88 (2.71–3.07), <0.01

CI = confidence interval; EMS = emergency medical service; OR = odds ratio; SBP = systolic blood pressure.

**Table 3 jcm-10-05355-t003:** Prehospital baseline characteristics and mortality–multivariate analysis.

Characteristic Total *n* = 64,320	1 Month Mortality	3 Month Mortality	12 Month Mortality
Mortality, *n* (%)	1599 (2.5)	2549 (4.0)	4898 (7.6)
	OR (95% CI), *p*-value	OR (95% CI), *p*-value	OR (95% CI), *p*-value
Age over 80 years	3.37 (3.06–3.73), <0.01	3.26 (3.01–3.53), <0.01	3.14 (2.97–3.33), <0.01
Male	1.16 (1.04–1.28), 0.01	1.17 (1.08–1.27), <0.01	1.2 (1.14–1.28), <0.01
Comorbidities			
Congestive Heart Failure	1.43 (1.26–1.62), <0.01	1.57 (1.43–1.73), <0.01	1.91 (1.79–2.05), <0.01
Hypertension	0.68 (0.61–0.75), <0.01	0.73 (0.67–0.79), <0.01	0.79 (0.75–0.84), <0.01
Diabetes Mellitus	1.22 (1.07–1.38), <0.01	1.19 (1.08–1.31), <0.01	1.28 (1.19–1.37), <0.01
Ischemic Heart Disease	0.9 (0.8–1), 0.05	0.95 (0.87–1.03), 0.22	1 (0.94–1.07), 0.96
Main symptom at Presentation			
Chest Pain with ECG Ischemic Changes	1.95 (1.71–2.23), <0.01	1.41 (1.245–1.61), <0.01	1.40 (1.29–1.54), <0.01
Dyspnea	2.55 (2.29–2.83), <0.01	2.50 (2.29–2.72), <0.01	2.51 (2.29–2.72), <0.01
SBP at Presentation			
First EMS SBP < 130 mm Hg	2.26 (2.01–2.53), <0.01	2.06 (1.88–2.26), <0.01	1.71 (1.6–1.83), <0.01
First EMS SBP ≥ 160 mm Hg	0.86 (0.75–0.98), 0.03	0.88 (0.8–0.98), 0.02	0.86 (0.8–0.92), <0.01

CI = confidence interval; EMS = emergency medical service; OR = odds ratio; SBP = systolic blood pressure.

## Data Availability

Data supporting reported results can be found emailing emy.carasso@gmail.com.

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
