# Peer review of "Clinical Predictors of Mortality in Prehospital Distress Calls by Emergency Medical Service Subscribers"

_jcm, 2021, doi:10.3390/jcm10225355_

Round 1

Reviewer 1 Report

This retrospective observational study used prehospital data from 1988 – 2018 to predict mortality. A total of 64320 patients requiring EMS (all comers) were followed up to 12 months. It was found that old age (>80 y) and SBP<130 mmHg were among the highest predictors. In contrast, SBP>160 mmHg was associated with reduced mortality.

  1. Systolic BP is well known as a significant and independent predictor of hospital morbidity and mortality. Previous studies have established a U-shaped relationship between SBP and short-term mortality; and implicated SBP alone may not be an adequate tool for risk stratification. Since SBP is the key focus of this study, a further analysis of SBP in increments of 10-mmHg from <90 up to >160 in association of clinical outcome will be more informative and of relevance.
  2. The study population seems to have high comorbidity of ischemic heart disease (IHD) (48.6%) at baseline. However, IHD was not associated with increased mortality at 1- and 3-month. Please provide some explanation.
  3. What is the definition of ECG ischemic changes in this study?
  4. Figure 1: The X-axis should be Time in (days), not months. Please correct.

Author Response

Reviewer 1

Comment 1:

Systolic BP is well known as a significant and independent predictor of hospital morbidity and mortality. Previous studies have established a U-shaped relationship between SBP and short-term mortality; and implicated SBP alone may not be an adequate tool for risk stratification.

Since SBP is the key focus of this study, a further analysis of SBP in increments of 10-mmHg from <90 up to >160 in association of clinical outcome will be more informative and of relevance.

Response to Comment 1:

We thank the reviewer for this important suggestion, as such we performed the SBP analysis as recommended in increments of 10-mmHg from <90 up to >160.  In the figure below we could find linear relationship between SBP and mortality with higher survival rate while SBP is higher and vice versa. We added the figure (Figure 2) to our manuscript and to the result section.

Please see the figures in the attachment.

Comment 2:

The study population seems to have high comorbidity of ischemic heart disease (IHD) (48.6%) at baseline. However, IHD was not associated with increased mortality at 1- and 3-month. Please provide some explanation.

Response to Comment 2:

EMS-SHL Telemedicine provide services to population with IHD.  We believe that patients with prior IHD who had such service also has close medical follow-up which may be the cause for this finding.

Comment 3:

What is the definition of ECG ischemic changes in this study?

Response to Comment 3:

We defined ECG ischemic changes as ST-segment elevation or ST-segment depression.  We added the definitions to the method section.

Comment 4:

Figure 1: The X-axis should be Time in (days), not months. Please correct.

Response to Comment 4:

We corrected the figure.

Reviewer 2 Report

Major comments:

  1. Pre-hospital clinical characteristics were minimal. They investigated the following clinical factors: (1) age, (2) sex, (3) history of congestive heart failure (CHF), (4) history of HTN, (5) history of DM, (6) history of ischemic heart disease (IHD), (7) SBP as performed and documented by the EMS team, but obesity or CKD is also essential for predicting future mortality. Would you please provide more clinical indicators, at least coronary risk factors, such a smoking history, CKD, or obesity (BMI)?
  2. As the authors mentioned in the limitation, this study excluded the general population of patients potentially admitted to the hospital due to life-threatening cardiovascular conditions. If so, the results are likely to be very different if the general population is included.
  3. The conclusion is very weak and vague. At the very least, please specify what was clarified by the results obtained from the logistic regression analysis.
  4. Would you please disclose and provide the Python code that the authors used for analysis?

Minor comments:

  1. Would you please provide numbers at risk below the Kaplan-Meier curve?
  2. Just in case, please check the P-value and digit numbers. Some of them in Figure 2 and 3 were odd (<0.005, <.005, 0.9(0.8-1) and so on).

Author Response

Reviewer 2

Comment 1:

Pre-hospital clinical characteristics were minimal. They investigated the following clinical factors: (1) age, (2) sex, (3) history of congestive heart failure (CHF), (4) history of HTN, (5) history of DM, (6) history of ischemic heart disease (IHD), (7) SBP as performed and documented by the EMS team, but obesity or CKD is also essential for predicting future mortality.  Would you please provide more clinical indicators, at least coronary risk factors, such a smoking history, CKD, or obesity (BMI)?

Response to Comment 1:

We agree with the reviewer that the risk factors mention above are important, unfortunately these DATA were not available from Shahal database, we added it to our limitations.

Furthermore, unfortunately, no data about other risk factors as CKD, obesity and smoking status have been included in our study, despite of increasing interest”.

Comment 2:

As the authors mentioned in the limitation, this study excluded the general population of patients potentially admitted to the hospital due to life-threatening cardiovascular conditions. If so, the results are likely to be very different if the general population is included.

Response to Comment 2:

We agree with the reviewer.  We analysed the population that were admitted to hospital via EMS.  There is a population bias and as such we added this to our limitation section.

“Additionally, there is a population bias, our study looked at a particular group of patients, those who could ‘afford’ a paid emergency plan such as EMS - SHL, and who were regarded ‘at risk’ for particular medical conditions, thus requiring a plan.  Our study excluded the population of patients potentially admitted to hospital due to life-threatening cardiovascular conditions by other transports ways.   

Comment 3:

The conclusion is very weak and vague. At the very least, please specify what was clarified by the results obtained from the logistic regression analysis.

Response to Comment 3:

We thank the reviewer for this suggestion we changed the conclusion.

We identified risk predictors for all-cause mortality in a large cohort of patients during prehospital EMS calls.  Age over 80 years, first EMS documented prehospital SBP<130 mmHg and dyspnea at presentation were the most profound risk predictors for short and long term mortality.  The current study demonstrates that in the prehospital EMS call settings, several parameters can be used to improve prioritization and management of high-risk patients.

Comment 4:

Would you please disclose and provide the Python code that the authors used for analysis?

Response to Comment 4:

We apologized for this mistake; we corrected the Statistical Software version that we used.

“MedCalc® Statistical Software version 20.014 (MedCalc Software Ltd, Ostend, Belgium; https://www.medcalc.org; 2021)”.

Comment 5:

Would you please provide numbers at risk below the Kaplan-Meier curve?

Response to Comment 5:

As recommended, we provided numbers at risk in the Kaplan-Meier curve.

Please see the figure in the attachment.

Comment 6:

Just in case, please check the P-value and digit numbers. Some of them in Table 2 and 3 were odd (<0.005, <.005, 0.9(0.8-1) and so on).

Response to Comment 6:

We re-checked the p-value in our tables and changed to 2 numbers following the digit.

Round 2

Reviewer 2 Report

  1. I appreciate your efforts on improving the manuscript. The authors changed Python software to MedCalc® Statistical Software. Normally, it is impossible to mistake the software name. 
  2. I did not find the paper novel at all. Please explain the detail of the new findings.

Author Response

Reviewer 2

Comment 1:

I appreciate your efforts on improving the manuscript. The authors changed Python software to MedCalc® Statistical Software. Normally, it is impossible to mistake the software name.

   Response to Comment 1:

Initial statistical analysis was done using Python, as the previous draft of the manuscript stated.  As more analyses and verification of previous findings were required by some of the co-authores after proof-reading the manuscript, we shifted to a full analysis using latest version of  MedCalc. The results in the manuscript are from MedCalc. We failed to update the stats section of the manuscript before submission to the journal.

Comment 2:

I did not find the paper novel at all. Please explain the detail of the new findings.

Response to Comment 2:

Our study included 64,000 patients, assessed and defined the real-world clinical parameters in the prehospital setting.  To our knowledge this is one of the biggest cohort trials in the literature.  Both long term and short outcome were evaluated based on the pre-hospital settings and specific symptoms and signs (such as dyspnea as a morbid complain and first BP in arrival were noted to have significant impact on mortality).  We believe these are important issues as they may benefit EMS teams globally 

Thank for considering our paper.